# Explaining the Space of Plans through Plan-Property Dependencies

**Rebecca Eifler,**[1] **Michael Cashmore,**[2] **Jörg Hoffmann,**[1]
**Daniele Magazzeni,**[2] **Marcel Steinmetz,** [1]

[1]Saarland University, Saarland Informatics Campus, Saarbrücken, Germany,
[2]King's College London, Department of Informatics, London, UK,
{lastname}@cs.uni-saarland.de, {firstname.lastname}@kcl.ac.uk

## Abstract

A key problem in explainable AI planning is to elucidate decision rationales. User questions in this context are often contrastive, taking the form "Why do A rather than B?". Answering such a question requires a statement about the space of possible plans. We propose to do so through plan-property dependencies, where plan properties are Boolean properties of plans the user is interested in, and dependencies are entailment relations in plan space. The answer to the above question then consists of those properties C entailed by B. We introduce a formal framework for such dependency analysis. We instantiate and operationalize that framework for the case of dependencies between goals in oversubscription planning. More powerful plan properties can be compiled into that special case. We show experimentally that, in a variety of benchmarks, the suggested analyses can be feasible and produce compact answers for human inspection.

## Introduction

Explainable AI (XAI) is concerned with making AI systems' decisions more lucid and thus trustworthy. AI planning is relevant to XAI as a decision-making methodology, model-based and thus suited to provide explanations in principle. Consequently, research on explainable AI planning (XAIP) has received increasing interest in recent years (e. g. (Seegebarth *et al.* 2012; Smith 2012; Langley *et al.* 2017; Fox *et al.* 2017; Chakraborti *et al.* 2017; 2019)).

A recent analysis (Miller 2019) of lessons to be learned for XAI from social sciences highlights that user questions are often *contrastive*. A question "Why this?" actually means "Why this *rather than something else* that I would expect?". To address such queries, explanatory systems should analyse alternative solutions, and support the user in understanding the consequences of the "something else" in question. AI planning fits well for this kind of analysis. Two prior works designed variants thereof (Fox *et al.* 2017; Miller 2018). The work by Fox et al. is the starting point of our work here.

Fox et al. suggest, given a plan $\pi$ and a user question "Why does $\pi$ start with action $A$ rather than $B$?", to generate a new plan $\pi'$ starting with $B$, and answer the question based on comparing the two plans: undesirable properties of $\pi'$ serve to explain the previous decision. While this idea is natural, a key weakness is that there may be differences between $\pi$ and $\pi'$ unrelated to the use of $A$ vs. $B$. Many comparison aspects (e. g. which other actions are used, or which "soft" objectives are satisfied) may be affected by arbitrary decisions in the planner's search.

Here we address the same kind of explanation problem, but we replace the *existential* answer generating a single alternative plan $\pi'$ with a *universal* answer determining shared properties of *all* possible such alternatives. In this way, the analysis we propose aims at explaining the space of possible plans, rather than pointing out examples.

Our proposed analysis works at the level of **plan properties**: Boolean functions on plans that capture aspects of plans the user cares about (whether or not the plan starts with a particular action, whether or not a particular soft objective is satisfied, etc). We assume that the set $P$ of plan properties of interest is given as part of the input.[1] Our analysis then determines the **dependencies** across plan properties, i. e., **plan-space entailments** which we define as follows. The "plan space" is the set $\Pi$ of candidate plans to be considered (canonically, the set of plans for an input planning task). A plan property $p$ **entails** another property $p'$ in $\Pi$ if every $\pi \in \Pi$ that satisfies $p$ also satisfies $p'$. A user question "Why does the current plan $\pi$ satisfy $p$ rather than $q$?" can then be answered in terms of the properties $q'$ not true in $\pi$ but entailed by $q$: things that will *necessarily* change when satisfying $q$.

Our approach also supports iterative planning, along the lines suggested by Smith (2012). Given a current plan $\pi \in \Pi$ and a user question "Why achieve $p$ rather than $q$?", if the consequences of $q$ are tolerable to the user, she may choose to enforce $q$, gradually narrowing the plan-candidate space $\Pi$.

We remark that our approach can be viewed as an intermediate between domain/task analysis (e. g. (Fox and Long 1998)), which our approach generalizes; and model checking applied to planning models, which our approach is an instance of (related to (Vaquero *et al.* 2013)).

Our contributions are as follows. We conceptualize the explainability problems we address, through a generic framework making minimal assumptions on the planning context (Section ). We instantiate the framework with goal-

---

[1]An interesting yet challenging question for future work is how one can automatically identify relevant plan properties.

fact conjunction dependencies in oversubscription planning (e. g. (Smith 2004; Domshlak and Mirkis 2015)), and devise analysis algorithms for that purpose (Section ). We show that more general plan properties – in particular, **action-set properties** – can be compiled into goal facts and thus into that analysis (Section ). We give an illustrative example (Section ), and we evaluate our techniques on international planning competition (IPC) benchmarks modified for oversubscription planning, and on IPC benchmarks extended with action-set properties (Section ). We find that, in a variety of benchmark studies, the suggested analyses can be feasible and produce compact answers for human inspection.

# Generic Framework

We assume some formalism defining planning **tasks** $\tau$. We do not need any assumptions about that formalism, except that it defines a concept of **plans** $\pi$, where that concept can again be arbitrary (action sequence/schedule/partial order/etc). Our definitions are relative to a set $\Pi$ of plans of interest. The canonical setup we have in mind is that where $\Pi$ is induced by $\tau$, e. g. as the set of action sequences applicable in the initial state, or as the set of plans that achieve a goal. It could also be useful in some cases though to focus the analysis on a small set of candidate plans listed as part of the input.

## Plan Properties and Property Entailment

Plan properties, in their most general form, are simply functions mapping a task and plan to a Boolean value indicating whether or not the property is satisfied:

**Definition 1** (Plan Property). *Denoting by $\mathcal{T}$ the set of all tasks and by $\mathcal{P}$ the set of all plans, a* plan property *is a partial function $p : \mathcal{T} \times \mathcal{P} \mapsto \{true, false\}$. Given a task $\tau$ and a set of plans $\Pi$, we say that $p$ is a plan property defined on $\tau$ and $\Pi$ if its domain includes $\{(\tau, \pi) \mid \pi \in \Pi\}$.*

Example plan properties are goal facts/goal formulas (true at end of plan?), temporal plan trajectory constraints, constraints on subsets of actions used/not used, deadlines, bounds on resource consumption, etc. We expect that, typically, $p$ will be computable in time polynomial in the size of its input (though that is not a requirement of our framework).

We assume a set $P$ of plan properties as part of our input. $P$ may be exponentially large in the size of its specification though. An example we will explore later is that where the user is interested in dependencies between subsets of a set $G$ of soft-goal facts. The set $P$ of interest then are the conjunctions $\phi$ over $G$ (functions checking whether $\phi$ is true at the end of a plan), but the input to our analysis specifies only $G$.

The kind of dependency our framework focuses on is entailment over plan properties, in the space of truth-value assignments induced by the plan-candidate set $\Pi$:

**Definition 2** ($\Pi$-Entailment). *Let $\tau$ be a task, $\Pi$ a set of plans, and $P$ a set of plan properties defined on $\tau$ and $\Pi$.*

*Let $\pi \in \Pi$. We identify $\pi$ with the truth-value assignment $\pi : P \mapsto \{true, false\}$ where $\pi(p) := p(\tau, \pi)$. We identify $\Pi$ with the set of such truth-value assignments. We say that*

$\pi$ **satisfies** $p$, written $\pi \models p$, if $\pi(p) = true$. We denote by $\mathcal{M}_{\Pi}(p) := \{\pi \mid \pi \in \Pi, \pi \models p\}$ the models of $p$.

*We say that $p$ $\Pi$-**entails** $q$, written $\Pi \models p \Rightarrow q$, if $\mathcal{M}_{\Pi}(p) \subseteq \mathcal{M}_{\Pi}(q)$. We say that $p$ and $q$ are $\Pi$-**equivalent**, written $\Pi \models p \Leftrightarrow q$, if $\mathcal{M}_{\Pi}(p) = \mathcal{M}_{\Pi}(q)$. We denote $[p]_{\Pi} := \{q \mid q \in P, \Pi \models p \Leftrightarrow q\}$.*

This definition essentially just views plans $\pi \in \Pi$ as truth-value assignments in the obvious manner. Entailment and equivalence over plan properties are then defined straightforwardly, with $\Pi$ in the role traditionally taken by a knowledge base that restricts the truth-value assignments under consideration. Observe that formulas over plan properties can be encoded as individual plan properties, so that defining $\Pi$-entailment over individual plan properties is enough to permit logical combinations thereof.

Importantly, the role of $\Pi$ as a knowledge base means that $\Pi$-entailment is more than standard entailment: the latter implies the former, but not vice versa. As a simple example, say the plan properties $P$ are propositional formulas $\phi$ over facts, evaluated at the end of the plan. Then $\phi \Rightarrow \psi$ implies that $\Pi \models \phi \Rightarrow \psi$, simply because any (plan-end) state that satisfies $\phi$ must satisfy $\psi$. But not vice versa: e. g. if facts $p, q$ are mutex in the task then $\Pi \models p \Rightarrow \neg q$. As a more motivating example, say the plan properties are soft goals (like having scientific observations in satellite planning) as well as resource preferences (like consuming at most a given amount of energy). Then entailments of interest can take the form $\Pi \models p \Rightarrow \neg(q_1 \wedge q_2 \wedge q_3)$ saying that we cannot have $p$ without foregoing either of $q_1$ or $q_2$ or $q_3$. Note that this is an entailment specific to $\Pi$, which may not hold in general (e. g. if cheaper actions are available, or if cheaper plans are admitted by removing some other hard goals). The identification of such specific entailments – specific to the space $\Pi$ of plans considered – is central to our framework.

## Plan-Space Explanations

Our plan-space explanations are based on the $\Pi$-entailment relation on $P$ given the knowledge base $\Pi$:

**Definition 3** (PDO, cPDO). *Let $\tau$ be a task, $\Pi$ a set of plans, and $P$ a set of plan properties defined on $\tau$ and $\Pi$.*

*The **plan-property dependency order (PDO)** for $\Pi$ and $P$ is the partial order $\Rightarrow_{\Pi}$ over the equivalence classes $[p]_{\Pi}$, where $[p]_{\Pi} \Rightarrow_{\Pi} [q]_{\Pi}$ iff $\Pi \models p \Rightarrow q$.*

*A **concrete PDO (cPDO)** replaces each equivalence class $[p]_{\Pi}$ with exactly one $p \in [p]_{\Pi}$.*

The PDO makes explicit how the plan properties $P$ depend on each other. For all contrastive user questions of the form "Why $r$ rather than $p$?", the answer can be directly extracted from the PDO, in terms of the properties entailed by $p$. For example, the answer may be "we cannot have $p$ without foregoing either of $q_1$ or $q_2$ or $q_3$".

However, the PDO and the answers it provides can be large. A concrete PDO can be a practical proxy if equivalence classes are large. Beyond that, it is clearly important to identify (i) more compact and/or (ii) more restricted plan-space explanations. We introduce variants of both here.

Regarding (i), in our concrete instantiation of this framework we use **subsumption** over $\Pi$-entailment relations, relying on an easy-to-test sufficient criterion for $\Pi$-entailment:

**Definition 4** (Dominant cPDO). *Let $\tau$ be a task, $\Pi$ a set of plans, and $P$ a set of plan properties defined on $\tau$ and $\Pi$.*

*Let $\Rightarrow_{suff} \subseteq P \times P$ be such that, if $p \Rightarrow_{suff} q$, then $\Pi \models p \Rightarrow q$. In a cPDO, we say that $p \Rightarrow_\Pi q$ **subsumes** $p' \Rightarrow_\Pi q'$ **given** $\Rightarrow_{suff}$ if $p' \Rightarrow_{suff} p$ and $q \Rightarrow_{suff} q'$.*

*A **dominant cPDO (dcPDO)** for $\Pi$ and $P$ given $\Rightarrow_{suff}$ is the subset of non-subsumed $p \Rightarrow_\Pi q$ in a cPDO.*

An entailment $p \Rightarrow_\Pi q$ subsumes another one $p' \Rightarrow_\Pi q'$ if its left-hand side is weaker ($p' \Rightarrow_{suff} p$) and its right-hand side is stronger ($q \Rightarrow_{suff} q'$): in this case, $p' \Rightarrow_\Pi q'$ follows from $p \Rightarrow_\Pi q$. A dominant cPDO thus selects only the strongest $\Pi$-entailments in a cPDO, as a more compact representation of the information present in that cPDO.

The role of $\Rightarrow_{suff}$ here is to qualify the amount of information we are allowed to use in identifying this compact representation. This is important because, if we show compacted information to a user, then the user should be able to de-compact this information – to obtain whichever information the user is actually interested in – effortlessly. A simple restriction is for $\Rightarrow_{suff}$ to be computable in polynomial time, but cognitive abilities may necessitate stronger restrictions. Here we will consider goal-fact conjunctions and disjunctions, and use the trivial $\Rightarrow_{suff}$ where larger conjunctions are stronger while larger disjunctions are weaker.

As a simple form of (ii) more restricted plan-space explanations, we will employ the restriction of focus to a predefined subset $D$ of dependencies of interest:

**Definition 5** (Restricted (dc)PDO). *Let $\tau$ be a task, $\Pi$ a set of plans, and $P$ a set of plan properties defined on $\tau$ and $\Pi$.*

*Let $D \subseteq P \times P$ be any binary relation on plan properties. Then a **(dc)PDO** for $D$ results from ignoring $\Pi$-entailments $\Pi \models p \Rightarrow q$ where $(p, q) \notin D$.*

Some words are in order regarding complexity. Testing $\Pi$-entailment encompasses the plan existence problem even for extremely simple plan properties (asking whether the plan achieves a fact $p$). This is exacerbated by the size of the PDO. Certainly, a (dc)PDO should ideally be computed offline, prior to interaction with a user.

## Goal Dependencies

We now instantiate our framework with a concrete use case: dependencies between goals in oversubscription planning, where the question addressed is which combinations of (soft) goals exclude which other combinations. In Section , we will show how to compile a more powerful plan property language into this special case.

### Planning Framework

Most of the techniques we introduce in what follows are applicable to a broad range of planning frameworks. Nevertheless, for a concrete exposition, henceforth we consider the *finite-domain representation (FDR)* framework (Bäckström and Nebel 1995; Helmert 2009), with finite-domain state variables as used in the Fast Downward system (Helmert 2006) on which our implementation is based.

An FDR task $\tau$ is a tuple $\tau = (V, A, c, I, G)$ where $V$ is the set of **variables**, $A$ is the set of **actions**, $c : A \mapsto \mathbb{R}_0^+$ is the action **cost** function, $I$ is the **initial state**, and $G$ is the **goal**. A **state**, in particular $I$, is a complete assignment to $V$; $G$ is a partial assignment to $V$; each action $a \in A$ has a **precondition** $pre_a$ and an effect $eff_a$, both partial assignments to $V$. We will refer to variable-value pairs $v = d$ as **facts**, and we will identify partial variable assignments with sets of facts. An action $a$ is **applicable** in a state $s$ if $pre_a \subseteq s$. The outcome state $s[[a]]$ is like $s$ except that $s[[a]](v) = eff_a(v)$ for those $v$ on which $eff_a$ is defined. The outcome state of an iteratively applicable action sequence $\pi$ is denoted $s[[\pi]]$.

We address an oversubscription variant of FDR, where an **oversubscription planning (OSP) task** is a tuple $\tau = (V, A, c, I, G, b)$ exactly like an FDR task but with an additional **cost bound** $b \in \mathbb{R}_0^+$. Intuitively, the goals $G$ are "soft", and the challenge is to achieve a maximally valuable subset of $G$ within the cost bound. OSP frameworks in the literature employ notions (e. g. goal-fact rewards) of what it means to be "maximally valuable". Here we assume instead that the user's preferences over the soft goals are difficult to specify and/or elicit, so that an in-depth characterization of the trade-offs between different goal sets – their dependencies – is of interest. In the terms of our framework, this means that the set $\Pi$ of **plans** is simply the set of all action sequences $\pi = \langle a_1, \ldots, a_n \rangle$ applicable in $I$ and where $\sum_{i=1}^n c(a_i) \leq b$. An analysis over suitable sets of properties $P$ and dependencies $D$ then yields the desired trade-off information.

### Plan Properties

The plan properties we consider here are characterized by propositional formulas over goals:

**Definition 6** (Goal Properties). *Let $\tau = (V, A, c, I, G, b)$ be an OSP task, and $\Pi$ its set of plans.*

*A **goal property** for $\tau$ is a function $p_\phi : \Pi \mapsto \{true, false\}$ where $\phi$ is a propositional formula over the atoms $G$, and $p_\phi(\pi) = true$ iff $\phi$ evaluates to true under the truth value assignment where $g \in G$ is true iff $g \in I[[\pi]]$.*

We identify goal properties $p_\phi$ with the characterizing formulas $\phi$. We consider a class of properties and dependencies identifying exclusions between goal conjunctions:

**Definition 7** (Goal Exclusion). *Let $\tau = (V, A, c, I, G, b)$ be an OSP task, and $\Pi$ its set of plans.*

*The **PDO for goal exclusion (PDO-GE)** is the PDO for $\Pi$, the property set $P^{\text{GE}} := \{\bigwedge_{a \in A} g \mid A \subseteq G\} \cup \{\neg \bigwedge_{g \in B} b \mid B \subseteq G\}$, and the dependency set $D^{\text{GE}} := \{(\bigwedge_{a \in A} a, \neg \bigwedge_{b \in B} b) \mid A \cap B = \emptyset\}$.*

We restrict focus to goal conjunctions and negations thereof, and we are interested only in implications of the form $\Pi \models \bigwedge_{a \in A} a \Rightarrow \neg \bigwedge_{b \in B} b$ stating that, if we achieve all of $A$, we have to forego at least one of $B$. The PDO-GE then explains to the user how exactly different goal subsets exclude each other, identifying the fine-grained trade-off.

Given the restriction to $D^{\text{GE}}$, the equivalence classes in the PDO-GE are singletons. Hence there is a unique cPDO-GE, that we identify with the PDO-GE itself.

For compacting the information presented to a user, we use the sufficient criterion for entailment where $\bigwedge_{a \in A'} g \Rightarrow_{suff} \bigwedge_{a \in A} a$ iff $A' \supseteq A$ and $\neg \bigwedge_{b \in B} b \Rightarrow_{suff} \neg \bigwedge_{b \in B'} g$ iff $B \subseteq B'$. The dominant PDO-GE thus selects the entailments with minimal left-hand side conjunctions excluding minimal right-hand side conjunctions.

## Computing the Dominant PDO-GE

The dominant PDO-GE can be read off the **minimal unsolvable goal subsets (MUGS)**, where $G' \subseteq G$ is a MUGS if $G'$ cannot be achieved but every $G'' \subsetneq G'$ can:

**Proposition 1** (PDO-GE from MUGS). *Let $\tau = (V, A, c, I, G, b)$ be an OSP task, and $\Pi$ its set of plans.*

*Then $\Pi \models \bigwedge_{a \in A} a \Rightarrow \neg \bigwedge_{b \in B} b$ is in the dominant PDO-GE if and only if $A \cup B$ is a MUGS.*

*Proof.* A $\Pi$-entailment $\Pi \models \bigwedge_{a \in A} a \Rightarrow \neg \bigwedge_{b \in B} b$ clearly holds iff $A \cup B$ is unsolvable. Dominant entailments in the PDO-GE result from set-inclusion minimal $A$ and $B$, corresponding to the set-inclusion minimality of MUGS. □

Our computational problem thus boils down to computing all MUGS. This can be done through a search over goal sets, that we refer to as **systematic weakening (SysW)**:

(1) the start node of the search is $G$;

(2) each search step selects an open node $G'$, calls a planner to test whether $G'$ is solvable in $\tau$, caches the result, and expands $G'$ if it is unsolvable;

(3) the children of a node $G'$ are those $G'' \subset G'$ where $|G''| = |G'| - 1$.

Upon termination, the MUGS are those nodes $G'$ all of whose children are solvable.

Dually, **systematic strenghtening (SysS)** starts from $\emptyset$, with search steps expanding solvable nodes, and children adding one more goal fact. Upon termination, the MUGS can be easily obtained from the unsolvable search nodes.

In both SysW and SysS, every goal set can be reached from the start node by permutations of the same goal-fact removal/addition steps. We avoid duplicate planner calls by caching. We give goal sets unique integer IDs, for fast cache lookup, and to fix the expansion order so that we always know whether or not we have generated a node before.

As a non-trivial search enhancement, we created synergy with recent nogood learning techniques, **conjunction learning** (Steinmetz and Hoffmann 2017b) and **trap learning** (Steinmetz and Hoffmann 2017a). These techniques refine dead-end detection methods (nogoods) based on the unsolvable states encountered in state space search on a planning task. As the children tasks in our searches are closely related to their parents, the refined nogoods are likely to be useful still. So we **transfer** the nogoods along search paths, resulting in iteratively stronger and stronger nogoods. For both conjunction learning and trap learning, the nogoods learned depend on the goal, so that only some of the nogoods remain valid for transfer in SysW where children remove goals. We designed simple methods to identify this nogood

subset, keeping track of goal dependencies in conjunction learning, and re-asserting trap validity in trap learning.

Yu et al. (2017) perform an analysis related to MUGS, to suggest goals to drop in oversubscribed situations. They address conditional temporal problems (a form of conditional temporal plans), and leverage previous conflict analysis methods in that area. It remains a question for future work whether such conflict analysis could inspire different analysis methods in our planning framework.

## Compilations into Goal Dependencies

The analysis of goal properties just described can be used to analyze more complex properties that can be compiled into goal facts. Given the well-known power of compilation in planning languages (e. g. (Gazen and Knoblock 1997; Nebel 2000; Edelkamp 2006; Palacios and Geffner 2009; Baier *et al.* 2009)), there is large potential in this idea. As an example, here we consider what we refer to as action-set properties:

**Definition 8** (Action-Set Properties). *Let $\tau = (V, A, c, I, G, b)$ be an OSP task, $\Pi$ its set of plans, and $A_1, \dots, A_n \subseteq A$.*

*An **action-set property** for $\tau$ and $A_1, \dots, A_n$ is a function $p_\phi : \Pi \mapsto \{true, false\}$ where $\phi$ is a propositional formula over the atoms $A_1, \dots, A_n$, and $p_\phi(\pi) = true$ iff $\phi$ evaluates to $true$ under the truth value assignment where $A_i$ is $true$ iff $\pi$ contains at least one action from $A_i$.*

As before, we identify action-set properties $p_\phi$ with the characterizing formulas $\phi$. Arguably, action-set properties are practically relevant. They allow to express things like "objective x is covered by satellite y", "route x is not used", "passengers x and y ride in the same vehicle", etc. At the same time, the simple syntax of action-set properties lends itself to effective compilation, as follows.

Given $\tau$, $\Pi$, and $A_1, \dots, A_n$ as in Definition 8, to obtain a compiled task $\tau'$

1) introduce Boolean flags $isUsed_i$ that are initially false and set to $true$ by any action from $A_i$;

2) introduce formula-evaluation state variables and actions evaluating each $p_\phi$ based on these (following (Gazen and Knoblock 1997; Nebel 2000)), setting Boolean flags $isTrue_\phi$ storing the outcome values;

3) introduce a separate 1) *planning phase* vs. 2) *formula-evaluation phase*, and a switch action allowing to go from 1) to 2).

Then the planning-phase prefixes in $\tau'$ are in one-to-one correspondence with $\Pi$, and given such a prefix $\pi$ the evaluation phase in $\tau'$ can achieve $isTrue_\phi$ iff $p_\phi(\pi) = true$.

Now say that we want to analyze the dependencies across a given set $P$ of action-set properties (e. g. possible undesirable consequences of not using route X). We are given $\tau$, $\Pi$, and $P$; we want to compute the PDO for property exclusion over $P$, i. e., the dependencies of the form $\Pi \models \bigwedge_{\phi \in A} \phi \Rightarrow \neg \bigwedge_{\psi \in B} \psi$. With the above, this can be done by instead computing the PDO-GE for $\tau'$ with goal set $\{isTrue_\phi \mid \phi \in P\}$, and identifying each $isTrue_\phi$ with $\phi$ in the outcome.

Clearly, similar compilation techniques can be used for much more powerful property languages. In a preliminary exploration, we implemented a compilation for LTL properties based on previous work (Edelkamp 2006; Baier *et al.* 2009). Our results indicate that this renders the PDO analysis infeasible computationally. It remains an open question how LTL properties can be addressed more effectively.

## An Illustrative Example

To illustrate our approach and the kind of explanations it provides, consider the IPC NoMystery domain, a classical transportation domain with fuel consumption. We consider the example task with two trucks and three packages as illustrated below. Fuel costs are indicated at road segments (initial fuel is 16 for $T_0$ and 7 for $T_1$). The packages are initially at $L_0$ (shown in blue); their goal locations are $L_4$, $L_3$, and $L_5$ (shown in red). We define three kinds of action-set properties for this domain: *uses* $T_i$ $(L_x, L_y)$ (truck $T_i$ drives at least once from $L_x$ to $L_y$ or vice versa); *doesn't use* $T_i$ $(L_x, L_y)$ (the opposite); *same truck* $P_x$ $P_y$ (both packages are delivered by the same truck). In our example task, we consider six instances of these properties: 1. uses $T_0$ $(L_2, L_3)$; 2. same truck $P_1$ $P_2$; 3. uses $T_0$ $(L_4, L_3)$; 4. same truck $P_2$ $P_0$; 5. doesn't use $T_0$ $(L_0, L_5)$; 6. uses $T_1$ $(L_5, L_4)$.

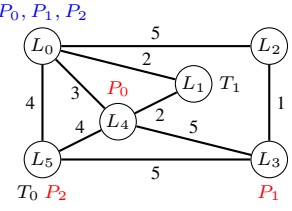

We fix the package destinations as hard goals, defining the set of plans $\Pi$ considered. Computing the MUGS over the six action-set properties using the algorithms previously described, it turns out there are seven minimal unsolvable subsets of these properties, each of size three.

Say now that the current plan uses $T_0$ only, and includes the action (drive T0 L5 L0). The user might ask *"Why don't you avoid the road $L_0 - L_5$, which has a lot of traffic at the moment?"*. Answering this question in terms of contrastive explanation, as previously discussed, corresponds to forcing property 5 to be satisfied. At the same time, the plan already satisfies properties 2 and 4. However, one of the MUGS is $\{2, 4, 5\}$, and hence the answer to the user question would be: *Because if you don't use that road, then you would not be able to deliver all packages using a single truck.*

## Experiments

We implemented our approach in Fast Downward (FD) (Helmert 2006). We evaluate it, in turn, on IPC benchmarks modified for oversubscription planning, and on a selection of IPC benchmarks extended with action-set properties.

In all experiments, the base planner called by our SysS and SysW algorithms on each search node employs $h^{\text{FF}}$ (Hoffmann and Nebel 2001) for search guidance. The experiments were run on a cluster of Intel E5-2660 machines running at 2.20 GHz, with time (memory) cut-offs of 30 minutes (4 GB).

## Oversubscription Planning

To evaluate our analysis of goal dependencies as per Section , we modified all optimal-planning STRIPS IPC domains up to IPC'18. Following Domshlak and Mirkis (2015), for each benchmark task we ran an optimal planner (A\* with $h^{\text{LM-cut}}$ (Helmert and Domshlak 2009)) to determine the optimal plan cost $C$, then obtained OSP tasks by setting the cost bound to $b = x * C$ where $x \in \{0.25, 0.5, 0.75\}$. Our benchmark set consists of 46 domains, and contains those tasks solved by the optimal planner, and where the number of goal facts is $< 32$. We extended conjunction learning (Steinmetz and Hoffmann 2017b) to deal with cost bounds, thus enabling nogood learning and transfer in SysS and SysW.

Figure 1 shows our data. Consider first the coverage data (leftmost two parts). To have some sort of measure of how computationally difficult our proposed analysis is, we use reference points from classical planning. First, the $h^{\text{LM-cut}}$ column gives coverage for A\* with $h^{\text{LM-cut}}$ run on the original IPC instance without a cost bound. This provides a comparison to solvable optimal planning. Second, the $h^C$ columns give coverage for search, with nogood learning, on the respective cost-bounded instances, when all goals must be achieved and thus the task is unsolvable. This provides a comparison to proving unsolvability, in the same situation where our approach computes all MUGS. It is expected that our algorithms, solving a more complex problem, will perform worse than the reference points.[2] The question is, *how much worse?*

As a short summary of the answer provided by Figure 1 to that question, compared to the $h^{\text{LM-cut}}$ reference point, taking the per-domain best of our four algorithm configurations, for $x = 0.25$ we get equal coverage in 36 of the 46 domains, and in that sense are "not much worse" than optimal planning. For larger cost bounds, the solvable goal subsets become larger, and accordingly our analysis becomes harder. For $x = 0.5$ we get equal coverage in 23 domains, for $x = 0.75$ in 13. The comparison to the $h^C$ proving-unsolvability reference points is qualitatively similar, with equal coverage in 38, 25, and 20 domains for $x = 0.25, 0.5, 0.75$ respectively. Overall, it seems fair to say that our analyses can be feasible in many cases, in the sense of not being more infeasible than the most closely related classical planning problems.

While comparing our algorithm configurations against each other is not our focus here, observe in the rightmost part of Figure 1 that both SysS and SysW suffer from larger cost bounds, but that is less so for SysW. This is because, for small cost bounds, solvable goal sets are small and thus SysS terminates earlier; while for large cost bounds, solvable goal sets are large and thus SysW terminates earlier. Conjunction learning ($h^C$ in the table) is moderately beneficial.

Consider finally the #MUGS part of Figure 1. Observe that, if the user asks a question "Why $r$ rather than $p$?", the answer are the properties entailed by $p$, represented here

---

[2]Indeed, the first reference point is an upper bound to our coverage, as only solved instances are included in our benchmark set; and the second reference point is an upper bound for SysW as it constitutes the first search node in that algorithm.

| domain | $h^{\text{LM-cut}}$ - | $h^C$ 0.25 | $h^C$ 0.5 | $h^C$ 0.75 | Cov0.25 SysS | SysS $h^C$ | SysW | SysW $h^C$ | Cov0.5 SysS | SysS $h^C$ | SysW | SysW $h^C$ | Cov0.75 SysS | SysS $h^C$ | SysW | SysW $h^C$ | avg 0.25 | avg 0.5 | avg 0.75 | max 0.25 | max 0.5 | max 0.75 | STF0.25 S | W | STF0.5 S | W | STF0.75 S | W |
|---|---|---|---|---|---|---|---|---|---|---|---|---|---|---|---|---|---|---|---|---|---|---|---|---|---|---|---|---|
| agricola (20) | 0 | - | - | - | - | - | - | - | - | - | - | - | - | - | - | - | - | - | - | - | - | - | - | - | - | - | - | - |
| airport (50) | 28 | 28 | 24 | 17 | 25 | 26 | 24 | **27** | 19 | **21** | 19 | **21** | **19** | 16 | **19** | 16 | 2.7 | 2.0 | 1.2 | 11 | 5 | 4 | **0.67** | 0.76 | 0.88 | **0.71** | **1.00** | **0.61** |
| barman (34) | 4 | 4 | 4 | 4 | 4 | 4 | 4 | 4 | 4 | 4 | 4 | 4 | **4** | 0 | 4 | **4** | 3.0 | 3.0 | 1.0 | 3 | 3 | 1 | **0.50** | 0.88 | 0.88 | 0.88 | - | - |
| blocks (35) | 28 | 28 | 28 | 28 | **28** | **28** | 27 | 28 | 23 | **27** | 21 | **27** | 18 | **24** | 17 | **26** | 7.6 | 10.8 | 14.1 | 39 | 30 | 57 | 0.19 | **0.97** | 0.39 | **0.93** | 0.78 | **0.72** |
| childsnack (20) | 0 | - | - | - | - | - | - | - | - | - | - | - | - | - | - | - | - | - | - | - | - | - | - | - | - | - | - | - |
| data-network (20) | 12 | 12 | 12 | 12 | 12 | 12 | 12 | 12 | 12 | 12 | 12 | 12 | 11 | **12** | 11 | **12** | 1.7 | 1.5 | 1.2 | 3 | 3 | 2 | 0.83 | **0.65** | 0.88 | **0.65** | 0.92 | **0.61** |
| depot (22) | 7 | 7 | 7 | 6 | 7 | 7 | 7 | 7 | 7 | 7 | 7 | 7 | 4 | 3 | 4 | 3 | 4.0 | 7.0 | 4.5 | 6 | 12 | 10 | **0.34** | 0.94 | **0.52** | 0.91 | **0.89** | 0.68 |
| driverlog (18) | 13 | 13 | 13 | 11 | 13 | 13 | 13 | 13 | 10 | 11 | 10 | **12** | 8 | **10** | 7 | **10** | 7.0 | 18.2 | 8.7 | 22 | 45 | 17 | **0.19** | 0.98 | **0.58** | 0.86 | **0.85** | 0.50 |
| elevators (50) | 40 | 40 | 40 | 35 | 40 | 40 | 40 | 40 | **40** | 37 | 38 | 37 | **35** | 26 | 31 | 26 | 3.9 | 4.9 | 3.2 | 8 | 13 | 8 | **0.37** | 0.94 | **0.67** | 0.89 | **0.92** | 0.71 |
| floortile (36) | 13 | 13 | 13 | 6 | 7 | 7 | 6 | **8** | 2 | 2 | 2 | 2 | 2 | 1 | 2 | 2 | 175.6 | 66.0 | 31.5 | 697 | 71 | 33 | **0.12** | 0.99 | **0.67** | 0.80 | **0.97** | 0.28 |
| freecell (80) | 15 | 15 | 15 | 15 | 15 | 15 | 15 | 15 | 15 | 15 | 15 | 15 | **14** | 13 | 13 | 13 | 4.0 | 4.7 | 3.4 | 4 | 6 | 5 | **0.31** | 0.94 | **0.60** | 0.94 | **0.88** | 0.76 |
| ged (20) | 15 | 15 | 15 | 11 | 15 | 15 | 15 | 15 | **15** | 10 | 10 | 10 | 10 | 7 | 10 | 7 | 9.2 | 38.7 | 12.5 | 18 | 101 | 38 | **0.23** | 0.90 | **0.47** | 0.80 | **0.58** | 0.70 |
| grid (5) | 2 | 2 | 2 | 2 | 2 | 2 | 2 | 2 | 2 | 2 | 2 | 2 | 2 | 2 | 2 | 2 | 1.5 | 1.5 | 1.0 | 2 | 2 | 1 | **0.81** | 0.69 | **0.81** | 0.69 | **1.00** | 0.56 |
| gripper (14) | 7 | 7 | 7 | 5 | 5 | 5 | 5 | 5 | 4 | 4 | 4 | 4 | **4** | 3 | **4** | 3 | 458.3 | 87.0 | 39.5 | 1820 | 252 | 120 | **0.21** | 0.98 | **0.65** | 0.88 | **0.96** | 0.46 |
| hiking (20) | 9 | 9 | 9 | 9 | 9 | 9 | 9 | 9 | 9 | 9 | 9 | 9 | 9 | 9 | 9 | 9 | 1.4 | 1.4 | 1.0 | 2 | 2 | 1 | **0.89** | 0.61 | **0.89** | 0.61 | **1.00** | 0.61 |
| logistics (60) | 26 | 26 | 26 | 20 | 24 | **26** | 21 | **26** | 15 | 19 | 14 | **20** | 12 | 13 | 12 | **15** | 6.3 | 6.0 | 2.9 | 25 | 22 | 7 | **0.31** | 0.95 | **0.68** | 0.84 | **0.90** | 0.63 |
| miconic (150) | 141 | 120 | 76 | 50 | **66** | **66** | 55 | 64 | **45** | 40 | 44 | 43 | **41** | 36 | 40 | 36 | 76.0 | 24.1 | 8.4 | 363 | 98 | 36 | **0.06** | 0.99 | **0.73** | 0.82 | **0.95** | 0.61 |
| movie (30) | 30 | 30 | 30 | 30 | 30 | 30 | 30 | 30 | 30 | 30 | 30 | 30 | 30 | 30 | 30 | 30 | 7.0 | 35.0 | 21.0 | 7 | 35 | 21 | **0.34** | 0.99 | **0.67** | 0.94 | **0.94** | 0.50 |
| mprime (35) | 22 | 22 | 22 | 22 | 22 | 22 | 22 | 22 | 22 | 22 | 22 | 22 | 22 | 22 | 22 | 22 | 1.3 | 1.2 | 1.2 | 2 | 2 | 2 | **0.90** | 0.59 | **0.92** | 0.59 | **0.93** | 0.59 |
| mystery (30) | 17 | 17 | 17 | 17 | 17 | 17 | 17 | 17 | 17 | 17 | 17 | 17 | 15 | 17 | 15 | 17 | 1.4 | 1.4 | 1.2 | 2 | 2 | 2 | **0.88** | 0.63 | **0.88** | 0.63 | **0.92** | 0.63 |
| nomystery (20) | 14 | 14 | 14 | 13 | 14 | 14 | 14 | 14 | 10 | **12** | 10 | **12** | 8 | 8 | 8 | 8 | 7.3 | 18.5 | 5.8 | 18 | 47 | 13 | **0.20** | 0.96 | **0.63** | 0.92 | **0.87** | 0.61 |
| openstacks (77) | 47 | 45 | 45 | 43 | **45** | **45** | 37 | 43 | **45** | 43 | 29 | 41 | **42** | **42** | 22 | 33 | 15.3 | 14.2 | 12.3 | 25 | 25 | 23 | **0.06** | 0.99 | 0.05 | 0.99 | **0.18** | 0.97 |
| org-syn (20) | 7 | 7 | 7 | 7 | 7 | 7 | 7 | 7 | 7 | 7 | 7 | 7 | 7 | 7 | 7 | 7 | 5.1 | 5.1 | 5.1 | 12 | 12 | 12 | **0.23** | 0.94 | **0.23** | 0.94 | **0.23** | 0.94 |
| org-syn-s (13) | 10 | 10 | 10 | 9 | **8** | **8** | 7 | **8** | **8** | **8** | 7 | **8** | **7** | 6 | 6 | 6 | 5.2 | 7.2 | 8.3 | 12 | 28 | 36 | **0.20** | 0.95 | **0.23** | 0.95 | **0.32** | 0.89 |
| parcprinter (26) | 24 | 20 | 20 | 20 | 10 | 10 | 10 | **14** | 10 | 10 | 10 | **14** | 10 | 10 | 10 | **12** | 3.8 | 8.2 | 5.0 | 14 | 24 | 10 | **0.44** | 0.98 | **0.61** | 0.95 | **0.72** | 0.85 |
| parking (40) | 5 | 5 | 5 | 1 | **5** | 5 | 4 | **5** | 0 | **1** | 0 | **1** | 0 | 0 | 0 | 0 | 36.8 | 31.0 | - | 79 | 31 | - | 0.02 | 0.99 | - | - | - | - |
| pathways (23) | 5 | 5 | 5 | 5 | 5 | 5 | 5 | 5 | **4** | 5 | 4 | **5** | **4** | 4 | 4 | 4 | 3.2 | 3.8 | 1.8 | 6 | 10 | 3 | **0.53** | 0.81 | 0.77 | 0.77 | **0.91** | 0.70 |
| pegsol (2) | 2 | 2 | 2 | 2 | 0 | 0 | **2** | **2** | 0 | 0 | **2** | **2** | 0 | 0 | **2** | **2** | 7.0 | 23.5 | 64.0 | 8 | 41 | 122 | - | - | - | - | - | - |
| pipesworld-nt (50) | 17 | 17 | 17 | 17 | 17 | 17 | 17 | 17 | **17** | 17 | 16 | **17** | **16** | 14 | **16** | 14 | 3.7 | 6.4 | 3.8 | 8 | 31 | 17 | **0.44** | 0.89 | **0.73** | 0.84 | **0.88** | 0.66 |
| pipesworld-t (50) | 12 | 12 | 12 | 11 | 12 | 12 | 12 | 12 | 11 | 11 | 11 | 11 | **9** | 11 | **9** | 10 | 3.6 | 5.0 | 3.7 | 7 | 15 | 12 | **0.43** | 0.94 | **0.67** | 0.87 | **0.90** | 0.63 |
| psr-small (50) | 49 | 49 | 49 | 49 | 48 | 48 | **49** | **49** | 47 | 47 | 48 | **49** | 46 | 46 | **48** | 48 | 3.7 | 2.7 | 2.0 | 20 | 13 | 9 | **0.76** | 0.63 | **0.94** | 0.55 | **0.97** | 0.47 |
| rovers (31) | 8 | 8 | 8 | 7 | 8 | 8 | 8 | 8 | 7 | 7 | 7 | 7 | **6** | 5 | 6 | 4 | 18.0 | 11.4 | 3.8 | 95 | 35 | 12 | **0.36** | 0.93 | **0.74** | 0.84 | **0.91** | 0.59 |
| satellite (19) | 7 | 7 | 7 | 6 | 7 | 7 | 7 | 7 | 6 | 6 | 6 | 7 | 4 | 5 | 4 | **6** | 5.6 | 26.9 | 14.7 | 7 | 76 | 36 | **0.19** | 0.97 | **0.49** | 0.94 | **0.88** | 0.73 |
| scanalyzer (40) | 23 | 21 | 21 | 13 | 9 | **15** | 9 | 13 | 9 | 9 | 9 | 9 | **9** | 5 | **9** | **9** | 20.9 | 36.7 | 31.2 | 46 | 103 | 43 | **0.25** | 0.99 | **0.53** | 0.86 | **0.75** | 0.83 |
| snake (17) | 7 | 7 | 7 | 4 | 6 | 6 | 6 | 6 | 3 | 3 | 3 | 3 | **3** | 1 | 2 | 1 | 10.5 | 21.0 | 44.3 | 16 | 27 | 77 | **0.13** | 0.92 | **0.32** | 0.86 | **0.58** | 0.73 |
| sokoban (50) | 50 | 50 | 49 | 41 | **50** | **50** | 49 | **50** | **46** | 43 | 45 | 43 | **40** | 30 | 40 | 28 | 6.6 | 4.1 | 1.8 | 56 | 36 | 10 | **0.60** | 0.85 | 0.86 | **0.71** | 0.95 | 0.51 |
| storage (30) | 15 | 15 | 15 | 15 | 15 | 15 | 15 | 15 | 15 | 15 | 15 | 15 | **15** | 14 | **15** | 14 | 3.6 | 3.7 | 2.1 | 10 | 10 | 5 | **0.62** | 0.81 | **0.85** | 0.75 | **0.98** | 0.57 |
| termes (20) | 6 | 6 | 4 | 1 | **6** | 6 | 5 | 6 | **5** | 1 | 1 | 2 | 1 | 0 | 0 | 0 | 3.2 | 2.6 | 3.0 | 8 | 6 | 3 | **0.37** | 0.72 | 0.53 | 0.53 | - | - |
| tetris (17) | 6 | 6 | 6 | 5 | **6** | 6 | 5 | 6 | **4** | 3 | 3 | 3 | **3** | 2 | **3** | 2 | 29.7 | 32.8 | 7.3 | 81 | 82 | 11 | **0.26** | 0.98 | **0.81** | 0.77 | **0.97** | 0.41 |
| tidybot (40) | 23 | 23 | 23 | 19 | 23 | 23 | 23 | 23 | **23** | 22 | 23 | 22 | 13 | 13 | 7 | **14** | 3.1 | 3.3 | 3.4 | 4 | 6 | 6 | **0.38** | 0.92 | **0.75** | 0.84 | 0.96 | 0.66 |
| tpp (30) | 7 | 7 | 7 | 6 | 7 | 7 | 7 | 7 | 6 | **7** | 6 | 6 | **6** | 5 | 6 | 5 | 4.1 | 8.9 | 4.2 | 9 | 25 | 11 | **0.43** | 0.86 | **0.67** | 0.88 | 0.96 | **0.66** |
| transport (70) | 23 | 23 | 23 | 22 | 23 | 23 | 23 | 23 | 23 | 23 | 23 | 23 | **23** | 22 | 22 | 22 | 3.5 | 3.7 | 2.1 | 5 | 10 | 6 | **0.43** | 0.91 | **0.59** | 0.88 | **0.73** | 0.69 |
| trucks (30) | 10 | 10 | 10 | 8 | 10 | **10** | 9 | 10 | 6 | **7** | 6 | 7 | **5** | 3 | 5 | **4** | 15.7 | 17.1 | 5.0 | 36 | 31 | 8 | **0.23** | 0.97 | **0.70** | 0.89 | **0.93** | 0.65 |
| visitall (14) | 14 | 13 | 13 | 13 | **13** | **13** | 10 | 10 | 9 | **10** | 8 | **10** | 6 | 6 | 7 | **8** | 97.4 | 111.6 | 46.5 | 307 | 380 | 150 | **0.20** | 0.93 | **0.41** | 0.90 | **0.79** | 0.75 |
| woodworking (35) | 29 | 25 | 25 | 25 | **23** | **23** | 12 | 15 | **9** | **9** | 5 | **9** | 5 | 5 | 5 | 5 | 267.3 | 95.0 | 16.8 | 1030 | 192 | 26 | **0.02** | 0.99 | **0.27** | 0.93 | **0.72** | 0.52 |
| zenotravel (20) | 13 | 13 | 12 | 9 | **13** | **13** | 12 | **13** | **9** | **9** | 8 | 9 | 8 | **9** | 8 | **9** | 10.4 | 4.0 | 2.6 | 36 | 6 | 4 | **0.36** | 0.94 | **0.67** | 0.89 | **0.87** | 0.66 |
| Sum (1583) | 862 | 828 | 776 | 670 | 733 | **740** | 688 | 732 | 630 | 624 | 592 | **636** | **556** | 517 | 523 | 528 | | | | | | | | | | | | |

Figure 1: Results on IPC benchmarks modified for oversubscription planning. Reference Points: related classical planning tasks (see text). Coverage: of our MUGS algorithms SysS and SysW, with vs. without conjunction learning $h^C$. #MUGS: average/maximum number of MUGS, indicating explanation size (see text). Search Tree Fraction: fraction of worst-case search tree explored. Best performance in each part shown in **boldface**. Cost bounds set to $x$ times optimal cost.

through the smallest conjunctions excluded by $p$. The number of such conjunctions is at most the number of MUGS. So #MUGS corresponds to worst-case answer/explanation size. As the data shows, that size is often small, of a scale that seems feasible for human inspection.

## Action-Set Properties

To evaluate the use of our framework with more complex plan properties, beyond goal facts, we experimented with the compilation of action-set properties as per Section . We selected four IPC domains for extension with action-set properties, namely NoMystery, Rovers, and TPP as considered in resource-constrained planning (Nakhost *et al.* 2012), where minimum resource requirements are known as per available problem generators; plus the Blocksworld as an intuitively rather differently structured domain. In all four domains, we use discrete resource consumption encoded into the STRIPS model, enabling the use of trap learning (Steinmetz and Hoffmann 2017a) which turns out to be highly beneficial here.

In Blocksworld, we include two gripper hands and the action-set properties ask whether a given gripper is used to pick up a given block, or to stack a given pair of blocks. In NoMystery, the properties are as in our illustrative example (Section ). In Rovers, the properties ask whether a given rover or camera is used for a given observation. In TPP,

they ask whether given road segments are used, and whether given goods are bought at given markets. In all cases, we vary the number of action-set properties between 1 and 10. We fix the original goal facts as hard goals, and we set the available resources to $x \in \{1.0, 1.5, 2.0\}$ times the minimum needed to allow for costlier plans satisfying some of the properties.

We created benchmark tasks with size parameters around the borderline of computational feasibility for our analyses, given our time/memory limits. In Blocksworld, we used 5 – 8 blocks; in NoMystery, our tasks have 2 trucks, 6 locations, and 4 – 7 packages; in Rovers, they have 2 rovers, 5 waypoints, and 4 – 7 science objectives; in TPP, we use 5 markets, 1 depot, and 4 – 7 goods. In all domains, we vary the number of goal facts (and associated objects) between 4 and 7. We create 10 base instances for each size-parameter setting, which are then modified for our experiments with different initial resource levels, and action-set properties to be considered.

To have some comparison measure for performance, again we use classical-planning reference points, based on $A^*$ with $h^{\text{LM-cut}}$, and on search with trap learning, respectively. We now run these reference points on tasks where all (original goal facts plus) action-set properties are hard goals. These tasks may be solvable (in which case $A^*$ with $h^{\text{LM-cut}}$ tends to be better) or unsolvable (in which case trap learning

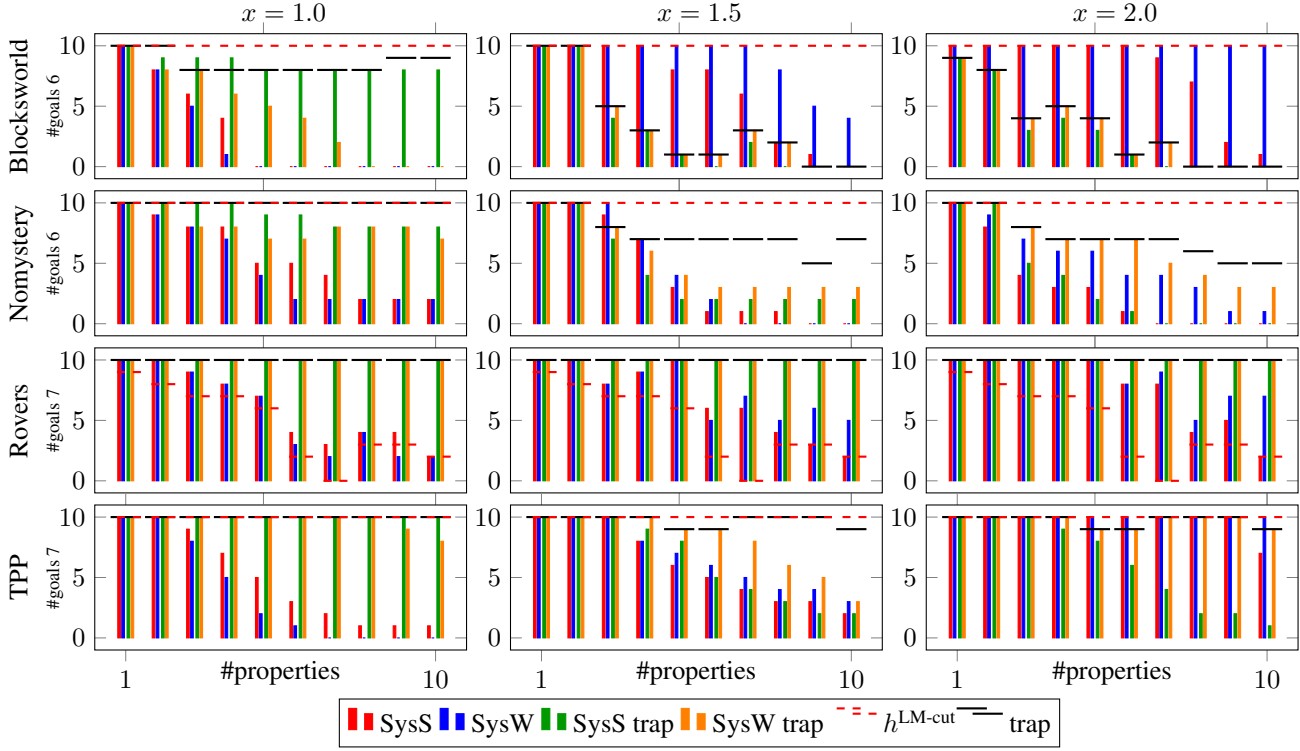

Figure 2: Coverage results on IPC benchmarks extended with action-set properties.

tends to be better). The configurations of our own algorithm are SysS and SysW as before, now with vs. without trap learning (and transfer).

Figure 2 shows the coverage data. For space reasons, we show only one row per domain, fixing the number of hard goals at the feasibility borderline. Smaller numbers of goal facts tend to be quite easy, larger ones mostly infeasible, with variance depending on the domain and algorithm.

## Conclusion

We introduced a framework for plan-space explanation via plan-property dependencies. We believe that the framework is useful conceptually as a problem formulation shaping a relevant part of XAIP. Our techniques for first instantiations of the framework exhibit reasonable performance in IPC benchmark studies. The computed explanations are often small and thus potentially feasible for human inspection.

In future work, the effectiveness of these explanations for human users remains to be evaluated in user studies. Another important question is how to address deeper "why" questions, asking for the reasons behind an entailment $\Pi \models p \Rightarrow q$. Possible ideas are to include additional properties into $P$, elucidating the causal chain between $p$ and $q$; or to find a minimal relaxation (superset) of the plan set $\Pi$ for which $p$ no longer entails $q$, thus elucidating the circumstances under which that entailment holds. Last but not least, of course our framework and algorithms can and should be

extended to richer planning frameworks and plan property languages.

## Acknowledgments

This material is based upon work supported by the Air Force Office of Scientific Research under award number FA9550-18-1-0245. Jörg Hoffmann's group has also received support by the German Research Foundation (DFG) as part of CRC 248 (see perspicuous-computing.science). Part of this work was performed while Jörg Hoffmann was visiting NASA Ames Research Center. We thank J. Benton, Minh Do, Jeremy Frank, and David Smith for insightful discussions.

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
