# OpenReview forum: "Explaining the Space of Plans through Plan-Property Dependencies"
_icaps-conference.org/ICAPS/2019/Workshop/XAIP — XAIP 2019_

### Official Review · AnonReviewer2 · 2019-05-07
**Provides a nice conceptual framework for contrastive explanation**

**Rating:** 4
**Confidence:** 2

**Review:**

This paper describes a general approach to contrastive explanation using plan-property dependencies. The key concept in this paper is plan-property depenency order, in which dependencies between properties over a set of plans are related. This provides a nice conceptual framework for asking contrastive questions over different *types* of properties, rather than just e.g. actions. The idea is instantiated as goal-dependencies, which is in turn implemented as oversubscription planning and the computational cost is evaluated on a large set of IPC benmarks. This demonstrates that the basic idea is computationally expensive, but not outrageously so!

Some more detailed comments:

- I like this idea of plan-space explanations, as outlined in Definition 3 and 4. It was quite a bit of work to get my head around this though because there are a few different concepts (with new syntax) leading up to this. I only really did that work because I was a reviewer and because this is so highly relevant to my own work. I think a nicely crafted example would immediately make this idea clear; even using an abstract example.

- I found Definition 7 difficult to follow. First, it would improve readability if there was some intuition given about when this is about. But more importantly, I could not understand the formalism here: what does A \subseteq G mean when A is a set of actions and G is a set of goals; what does the generalised conjunction mean to conjoin a goal given an action; and what is the set B referring to?

- I found the remainder of the section then a bit challenging and didn't understand parts of it because I didn't really know what PDO-GE was. I managed to recover into the compilations section.

- The illustrative example does a nice job of making the concepts in the former sections more concrete, but I did find the jump from the MUGS to the textual explanation quite large. Can you just add a brief example or language template from how to get from {2,4,5} to the text?

---

### Official Review · AnonReviewer1 · 2019-05-09
**Nice work formalizing the response to the "Why?" question to plans**

**Rating:** 3
**Confidence:** 3

**Review:**

The paper proposes an interesting framework for explaining plans in terms of binary plan properties.

> Description versus Explanation: My biggest criticism of the framework would be that it is an explanation framework without any consideration of the explainee in it! What is the model of the person who is receiving the explanation? How do they evaluate plans? Are they optimal planners? Can they arrive at the same plan space given a problem? Are they able to evaluate the entailment property completely and accurately?

---> 1. What is the definition of an explanation in this framework?
---> 2. Under what assumptions of the explainee model is this framework capable of producing a valid explanation?

Without these key definitions in the framework, I would say that the proposed work is one of plan description rather than explanation. The need for evaluations is noted only as an afterthought. This is worrying, especially since the framework does not account for the user model at all. As such, I am not sure who has been helped with explanations of this form. I would like to see more discussion on what the evaluation plans are, with respect to the design of experiments that evaluate not the descriptive power but the explanatory power of the framework.

> This also raises the issue that the set of properties are inputs to the algorithm. I see this as an artifact of not having the explainee model in the framework. I think the set of plan properties would follow directly from a complete framework that accounts for the explainee, either from their mental model of the task or from the questions asked. Interestingly, recent work [Sreedharan et al. IJCAI 2018] has looked at estimating this model (in a different context) and answer such why questions.

other: I am not sure that using coverage is an apples-to-apples comparison. Is there a reason to generate the explanation at planning-time? Usually, it is in response to the question (can potentially re-use search data).

other: I would strongly suggest merging (and expanding) the example at the end with the section where plan properties and entailment are introduced. It would be useful to carry this example through in following sections as well. I had a hard time going back and forth making sense of the presentation. Having a running example would really help in readability.

minor: section number depth is probably set to zero, all the section are missing.

---

> ### Author Response · Authors · 2019-05-09
> **Points well taken, in our view these things are future work, but we will try to add discussions**
>
> Thanks a lot for your comments!
>
> In our view, what we present the paper is the first step into a long research line (for which we have continued funding).
>
> The techniques proposed here are, in our view, the prerequisite for being able to answer questions about the space of plans at all, in a meaningful way. A lot is needed to build on this basis and address deeper why questions, model the user, derive interesting plan properties automatically.
>
> We're very much looking forward to discussing this work at the workshop, should it be accepted.

---

> > ### Author Response · Authors · 2019-05-09
> > **ps.**
> >
> > This was in the title of my previous comment, but let me be explicit:
> >
> > Your points regarding what's missing and reg the write-up are very well taken, thank you very much. We will make an effort to improve, and add discussions, for the final workshop version.

---

> > > ### Comment · AnonReviewer1 · 2019-05-14
> > > **completeness**
> > >
> > > Definitely. I think it's a nice framework, though I would still like q1 and q2 to be tackled for the framework to be complete. At the time of Definition 3, an "explanation" is not a defined concept, and the paper mentions that explanations are "based on \Pi-entailment" and that a plan-property dependency order can be used to explain. If the latter is equivalent to an explanation, then the authors might as well mention that explicitly.
> > >
> > > As it stands it is a great way of looking at plan space descriptions. As a framework for explanations, I think the concept of an explanation must be defined for completeness -- i.e. I was looking for an answer to the question: What is an explanation in this framework and why PDO counts as one?

---

### Official Review · AnonReviewer5 · 2019-05-15
**Excellent paper on universal answers to contrastive questions**

**Rating:** 5
**Confidence:** 2

**Review:**

A main idea of the paper, that a contrastive explanation should be answered 'universally' rather than 'existentially' is extremely interesting and novel.

The capability of this approach to explain how different goal subsets exclude each other is powerful because in any reasonably complex domain, this is could be unknown to the user. Such an understanding could also be helpful for other goal reasoning components if the planner is part of a larger cognitive system.

The evaluation is quite strong, and the experimental setup is described in detail. For Figure 2, it seems that the only analysis of the results is the last sentence before the conclusion, that " Smaller numbers of goal facts tend to be quite easy, larger ones mostly infeasible, with variance depending on the domain and algorithm. ". Is this the only interesting analysis of this figure? Even though space is tight, I would have appreciated more analysis here.

The paper is highly relevant to the workshop.

minor comments:
	* Section numbers throughout the document are missing

---

### Comment · Program_Chairs · 2019-05-23
**verbalizing plans**

I was thinking about the plan verbalization work from Manuela Veloso's group -- e.g. https://www.ijcai.org/Proceedings/16/Papers/127.pdf -- and realized there may be some interesting connections here. Granted these are different kinds of plans, motion plans versus plans in general, but...  I wonder if an approach like in this paper can be a better alternative to the template approach in the verbalization work. For example, a PDO versus a concrete PDO could help with "specificity" while planning models do have a rich literature of abstractions that can be readily leveraged in this framework. I guess the point is if it is useful automating the templatization using plan properties. Maybe something to think about...

More (recent) details here if you are interested:
[1] http://www.cs.cmu.edu/~mmv/papers/18gcai-PereraVeloso.pdf
[2] (data you can use) http://www.cs.cmu.edu/~vdperera/paper/vdperera_thesis.pdf (check appendix)

---

> ### Author Response · Authors · 2019-05-23
> **thanks!**
>
> wow, thanks for the interesting and helpful comment!

---

### Decision · Program_Chairs · 2019-05-15

**Decision:**

Accept

**Comment:**

The reviewers agree to accept. Please address all review criticism as best possible for the final paper version and its presentation at the workshop. Looking forward to discuss your work at the workshop!